# Evidence for RNA or protein transport from somatic tissues to the male reproductive tract in mouse

Vera Rinaldi[1], Kathleen Messemer[2,3], Kathleen Desevin[1], Fengyun Sun[1], Bethany C Berry[1], Shweta Kukreja[1], Andrew R Tapper[4,5], Amy J Wagers[2,3,6], Oliver J Rando[1]*

[1]Department of Biochemistry and Molecular Biotechnology, University of Massachusetts Chan Medical School, Worcester, United States; [2]Section on Islet Cell and Regenerative Biology, Joslin Diabetes Center, Boston, United States; [3]Department of Stem Cell and Regenerative Biology, Harvard University and Harvard Stem Cell Institute, Cambridge, United States; [4]Department of Neurobiology, University of Massachusetts Chan Medical School, Worcester, United States; [5]Brudnick Neuropsychiatric Research Institute, University of Massachusetts Chan Medical School, Worcester, United States; [6]Paul F. Glenn Center for the Biology of Aging, Harvard Medical School, Boston, United States

*For correspondence:
Oliver.Rando@umassmed.edu

**Abstract** The development of tools to manipulate the mouse genome, including knockout and transgenic technology, has revolutionized our ability to explore gene function in mammals. Moreover, for genes that are expressed in multiple tissues or at multiple stages of development, the use of tissue-specific expression of the Cre recombinase allows gene function to be perturbed in specific cell types and/or at specific times. However, it is well known that putative tissue-specific promoters often drive unanticipated 'off-target' expression. In our efforts to explore the biology of the male reproductive tract, we unexpectedly found that expression of Cre in the central nervous system resulted in recombination in the epididymis, a tissue where sperm mature for ~1–2 weeks following the completion of testicular development. Remarkably, we not only observed reporter expression in the epididymis when Cre expression was driven from neuron-specific transgenes, but also when Cre expression in the brain was induced from an AAV vector carrying a Cre expression construct. A surprisingly wide range of Cre drivers – including six different neuronal promoters as well as the adipose-specific *Adipoq* Cre promoter – exhibited off-target recombination in the epididymis, with a subset of drivers also exhibiting unexpected activity in other tissues such as the reproductive accessory glands. Using a combination of parabiosis and serum transfer experiments, we find evidence supporting the hypothesis that Cre may be trafficked from its cell of origin to the epididymis through the circulatory system. Together, our findings should motivate caution when interpreting conditional alleles, and suggest the exciting possibility of inter-tissue RNA or protein trafficking in modulation of reproductive biology.

## Editor's evaluation

This manuscript reports data consistent with a new and unanticipated phenomenon: that Cre or its mRNA may be transmitted between tissues in the mouse and that the male reproductive tract (epididymis) appears to be the most common target of such transported molecules. The data serve as a timely warning to mouse researchers about an unexpected complication of Cre-mediated gene manipulation.

## Introduction

The transition from unicellular life to multicellular life allowed organisms to delegate tasks to cells which could then specialize rather than being required to carry out all the functions of the organism. The disparate functions of different cell types are coordinated by a wide array of mechanisms, including direct cell-cell contacts within a tissue, shared access to nutrient pools, and both local (paracrine) and longer-range trafficking of small molecule and peptide signals. These signals often carry information about global organismal states such as hunger/satiety (insulin, leptin, etc.), reproductive status (testosterone, estrogen, etc.), stress (cortisol, adrenaline, etc.), and a multitude of other relatively blunt summaries of organismal status.

In addition to such coarse-grained signals of organismal physiology, it is increasingly understood that more detailed molecular signals can be communicated between distant tissues. This is particularly clear in the well-documented cases of systemic RNA trafficking seen in worms and plants. For instance, in *Caenorhabditis elegans*, introduction of double stranded RNA either by injection or by feeding with dsRNA-expressing bacteria leads to organism-wide silencing of homologous genes (*Fire et al., 1998*; *Timmons and Fire, 1998*), demonstrating that RNAs can traffic from the gut or hypodermis throughout the body, with sequence-specific regulatory consequences in the recipient tissue. Moreover, expression of RNAi triggers in neurons has been shown to silence target genes in the germline (*Devanapally et al., 2015*), again demonstrating trafficking of relatively precise RNA signals between tissues. Systemic RNA trafficking is also well described in plants, as various means of introducing dsRNA into aerial tissues can drive sequence-specific silencing in roots (*Palauqui et al., 1997*; *Voinnet et al., 1998*). These examples of systemic RNA trafficking highlight the potential for more precise molecular signaling between tissues than achievable via the relatively blunt information content available with hormonal signals.

In mammals, many studies have reported RNA or protein trafficking between different cell types within the same tissue (*van Niel et al., 2018*), or between distant tissues (*Thomou et al., 2017*; *Poggio et al., 2019*). Although this is well documented for several somatic tissues, in the unique case of the germline there are only sporadic reports of RNA trafficking from distant tissues to germ cells themselves (*Cossetti et al., 2014*; *O'Brien et al., 2020*; *Conine and Rando, 2022*). That said, it is worth noting that gametogenesis is supported by a wide variety of somatic cells and accessory tissues, and these could serve as intermediaries between distal tissues and developing gametes. These reproductive support cells/tissues include testicular cell populations such as the Sertoli cells that act as the spermatogonial stem cell niche and broadly support testicular spermatogenesis, as well as other tissues such as the epididymis, seminal vesicle, and prostate that contribute to sperm maturation and delivery to the female reproductive tract. Given the key roles of these tissues in sperm development and maturation, we were motivated to explore the potential for inter-tissue signaling to play a role in modulating reproductive tract physiology.

Here, we leverage the exquisite sensitivity of Cre-dependent recombination to identify inter-tissue communication pathways in the adult mouse, focusing on the male reproductive tract. We find that expression of Cre in multiple distant tissues, most notably including the central nervous system (CNS), results in expression of a Cre-dependent reporter gene in the epididymis. Reporter activity is observed following Cre expression in the CNS induced via adeno-associated viral (AAV) transduction, as well as in animals carrying Cre under the control of a variety of tissue-specific promoters. Parabiosis (in which two animals share a conjoined circulatory system) and serum transfer studies suggest that this 'off-target' reporter activity likely results from Cre trafficking from distant tissues to the epididymis via the circulatory system. Together, our findings highlight the epididymis as a surprisingly common site for unanticipated off-target effects in conditional knockout studies, and establish a novel model for inter-tissue communication in mammals.

## Results

Throughout this study, we utilize the Ai14D reporter mouse strain (*Madisen et al., 2010*; *Kauffman et al., 2018*) to assay Cre activity throughout the body (*Figure 1A*). This reporter carries a transgene at the 'safe harbor' *Rosa26* locus, with a ubiquitously expressed *CAG* promoter driving expression of a transgene with a 3× STOP cassette flanked by two *LoxP* sites, followed by a fluorescent reporter gene encoding tdTomato. Cre recombinase excises the STOP cassette, resulting in robust tdTomato

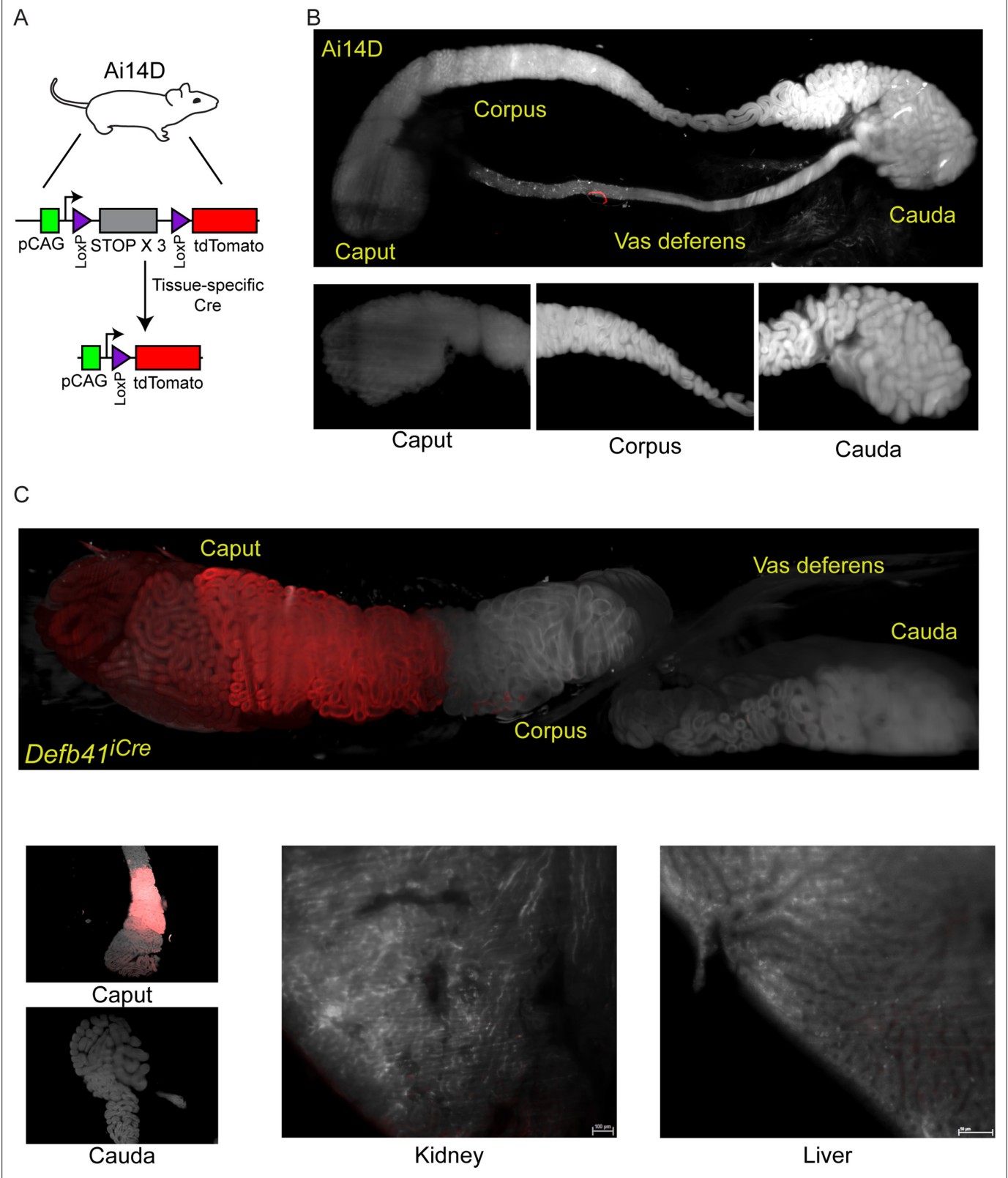

**Figure 1.** Analysis of Cre-dependent reporter gene expression in the murine epididymis. (**A**) Schematic of the Cre-dependent reporter present in the Ai14D strain. Reporter cassette, inserted at the safe harbor *Rosa26* locus, carries a strong p*CAG* promoter followed by a LoxP-flanked STOP cassette with three repeated in frame sequences. In the absence of Cre activity, transcription through this construct leads to transcriptional termination and no reporter activity. Following Cre-dependent excision of the LoxP cassette, the p*CAG* promoter is juxtaposed adjacent to a sequence encoding

*Figure 1 continued on next page*

*Figure 1 continued*

a tandem Tomato (tdTomato) fluorescent protein, resulting in robust expression of Tomato in tissues expressing Cre. (**B**) Lightsheet imaging of the mouse epididymis in the negative control Ai14D background. Following animal sacrifice, tissues were cleared for imaging according to a modified CLARITY protocol (Materials and methods). The entire epididymis, along with a large section of the vas deferens, was then imaged using the LaVision Biotec lightsheet microscope. Regions corresponding to the proximal (caput), middle (corpus), and distal (cauda) epididymis, and the vas deferens, are indicated on the image of the entire tissue sample. Insets show representative 2D slices of the indicated anatomical regions. (**C**) Positive control showing robust Tomato expression driven by the caput-specific *Defb41* Cre driver (***Björkgren et al., 2012***). Panels show whole tissue image and representative sections as described in panel (**B**), along with representative sections of the kidney and liver, as indicated. See also ***Figure 1—figure supplement 1***.

The online version of this article includes the following figure supplement(s) for figure 1:

**Figure supplement 1.** Histology of Ai14D; *Defb41^Cre* double transgenic.

(hereafter, Tomato) expression in Cre-expressing cells. For the majority of the experiments described below, Tomato expression is assayed in tissues cleared using a modified CLARITY (***Chung and Deisseroth, 2013***; ***Cronan et al., 2015***) protocol, and visualized using lightsheet microscopy to survey entire tissues. ***Figure 1B*** shows a typical negative control image of an epididymis taken from an Ai14D animal lacking any Cre expression. The larger panel includes the caput, corpus, and cauda epididymis along with a large section of the vas deferens, while smaller inset images show representative cross sections of the indicated regions of the epididymis.

As a positive control, we crossed the Ai14D animal to a mouse strain bearing the epididymis-specific *Defb41* iCre driver (***Björkgren et al., 2012***). ***Figure 1C*** shows Tomato expression throughout the epididymis of these double transgenic animals. Consistent with the distribution of *Defb41* mRNA expression observed in surveys of epididymal gene expression (***Johnston et al., 2005***; ***Rinaldi et al., 2020***), we find that *Defb41^iCre* drives reporter expression in a highly localized manner in the proximal (caput) epididymis. This segment-delimited expression is readily apparent in images of the whole epididymis, while cross sections show that Tomato expression is confined to the principal cells (but not basal or other interstitial cells) of the epididymal epithelium (***Figure 1C***, ***Figure 1—figure supplement 1***). Tomato expression is notably absent from the sperm-filled lumen. Together, these images provide representative positive and negative controls illustrating the utility of the Ai14D strain as a sensitive and specific reporter for Cre activity.

## Neuronal Cre expression results in reporter activity in the cauda epididymis

Motivated by observations in *C. elegans* that double-stranded RNA expression in neurons can result in siRNA activity in the germline (***Devanapally et al., 2015***), we set out to explore the potential for neuronal Cre expression in mice to drive cell non-autonomous Cre activity elsewhere in the body. In two series of experiments, Cre expression was induced in the mouse brain via stereotaxic delivery of AAV constructs engineered to drive Cre expression in transduced cells (***Figure 2A***). In an initial series of experiments, we utilized an AAV2-retro-Ef1a-Cre construct introduced into the prefrontal cortex (***Figure 2—figure supplement 1***), while follow-up studies targeted the lateral somatosensory cortex, caudate putamen, and nucleus accumbens using an AAV9-pCMV-Cre construct (***Figure 2B***). Remarkably, examination of the epididymis in both cases revealed robust Tomato expression in the cauda (distal) epididymis and in the vas deferens (***Figure 2C***). Tomato expression was reproducibly observed in all epididymis samples isolated from ten AAV-transduced animals, including three animals injected in the prefrontal cortex (***Figure 2—figure supplement 1***) as well as seven animals injected in the lateral somatosensory cortex, caudate putamen, and nucleus accumbens (***Figure 2C***, ***Figure 2—figure supplement 2***). As in the case of the *Defb41^iCre* driver, closer examination of our imaging data showed that Tomato expression was confined to the epididymal epithelium (inset, ***Figure 2C***, ***Figure 2—figure supplement 2***), with no detectable expression in the sperm filling the epididymal lumen (see below).

## Neuronal Cre expression driven from transgenes also induces recombination in the epididymis

An exciting potential explanation for the ability of neuronal Cre expression to drive recombination in the epididymis would be that Cre RNA or protein synthesized in neurons is somehow trafficked to the

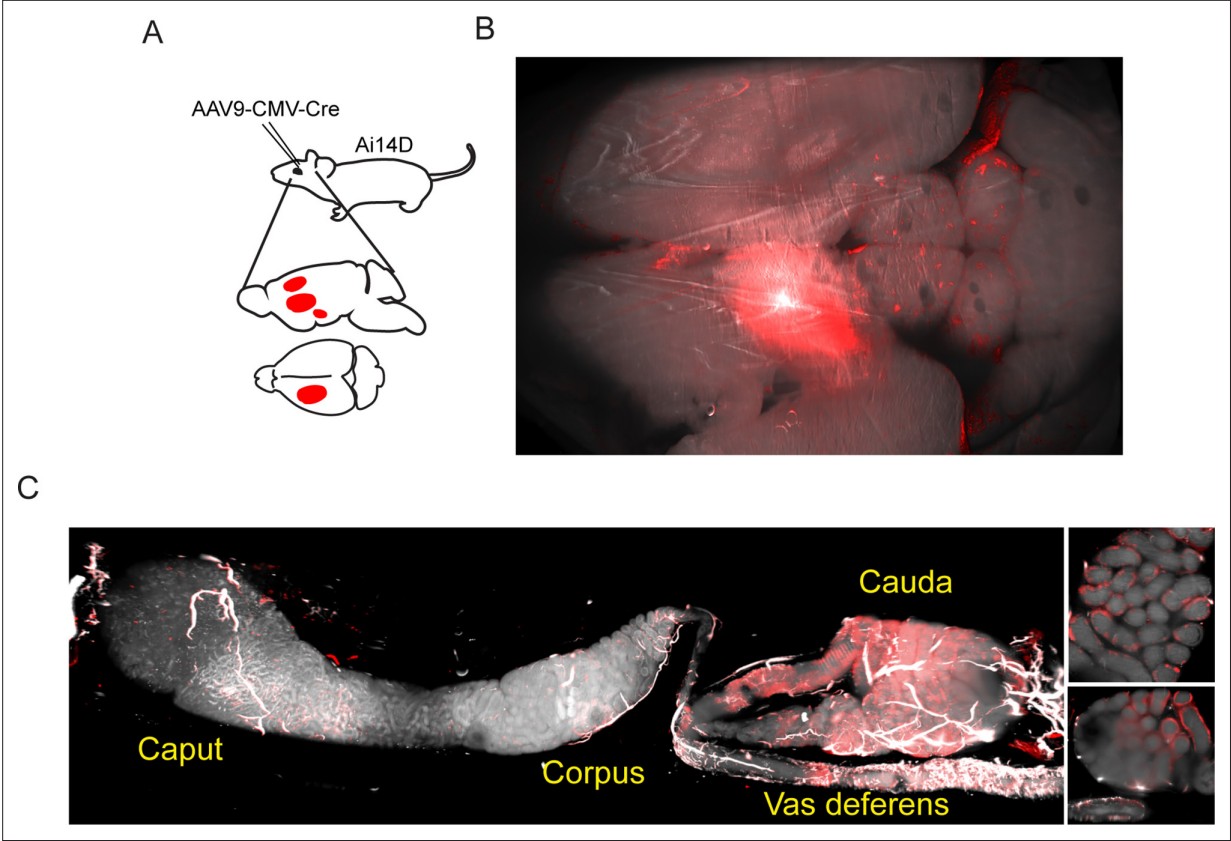

**Figure 2.** Adeno-associated viral (AAV)-mediated Cre expression in the brain drives reporter expression in the epididymis. (**A**) Experimental schematic showing stereotactically guided injection of AAV into the brain of Ai14D reporter animals to drive localized Cre-Lox recombination in the brain. Fifteen or 25 days post injection (depending on the experiment), mice were sacrificed and subject to tissue clearing and analysis by lightsheet microscopy. (**B**) Tomato reporter activity in the brain of typical AAV-injected animal. Panel shows maximum intensity projection for the entire mouse brain, viewed from above. (**C**) Images of the epididymis, as in *Figure 1B–C*. Right panels show two distinct 2D slices of the cauda epididymis, highlighting the Tomato-positive rim surrounding the Tomato-negative lumen. See also *Figure 2—figure supplements 1–2*.

The online version of this article includes the following figure supplement(s) for figure 2:

**Figure supplement 1.** Additional examples of Tomato-positive epididymis samples from reporter animals subject to adeno-associated viral (AAV)-mediated Cre expression in the central nervous system (CNS).

**Figure supplement 2.** Fluorescence microscopy images of epididymis from reporter animals subject to adeno-associated viral ( AAV)-mediated Cre expression in the central nervous system (CNS).

male reproductive tract. However, the use of viral transduction to induce Cre expression leaves open the possibility that viral particles entering the circulation could plausibly exhibit unanticipated tropism for the epididymis.

We therefore turned to an orthogonal method for driving Cre expression in neuronal subpopulations by crossing Ai14D mice to various driver lines expressing Cre under the control of cell type-specific promoters (*Figure 3A*). We illustrate this approach using *Slc32a1^{Cre}* mice that express Cre under the control of the vesicular GABA transporter (*Slc32a1*, or *Vgat*) promoter, which drives Cre expression in inhibitory GABAergic neurons in the CNS. Consistent with expectations, we observed widespread Tomato expression in the brain of double transgenic animals, with no detectable expression in the liver, kidney, intestine, or testis (*Figure 3B*). Remarkably, similar to the reporter expression observed following viral induction of neuronal Cre expression (*Figure 2*), Cre expression from the *Slc32a1* transgene also resulted in robust epididymal Tomato expression in Ai14D; *Slc32a1^{Cre}* double transgenic animals (*Figure 3C*). Closer inspection of double transgenic epididymis by microscopy revealed Tomato expression exclusively in the principal cells of the epididymal epithelium (*Figure 3D*), and we further confirmed the presence of Tomato-positive principal cells by FACS (*Figure 3—figure supplement 1*). Intriguingly, we observed patchy Tomato expression in any given histological cross

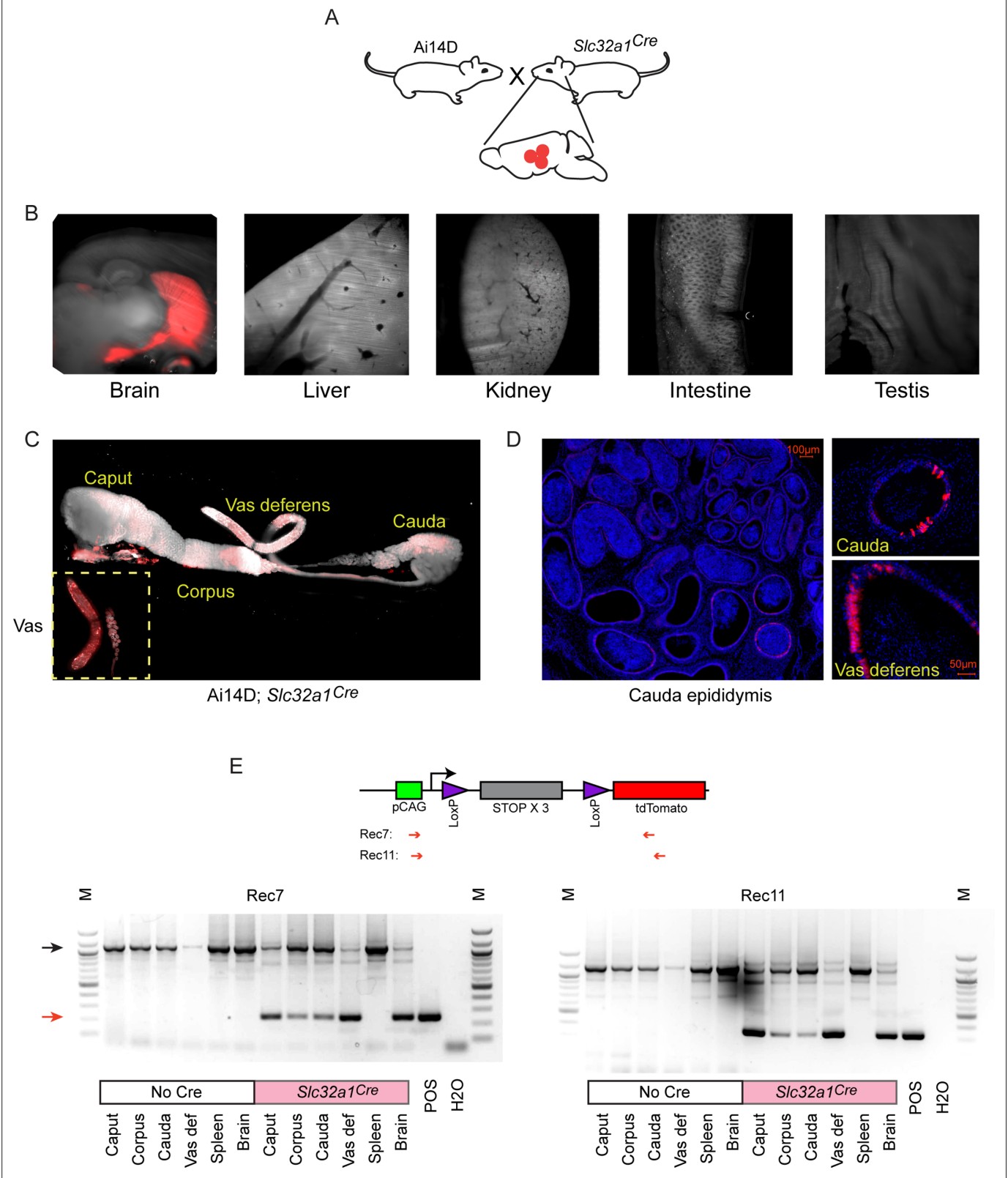

**Figure 3.** Cre expression in GABAergic neurons drives reporter expression in the epididymis. (**A**) Schematic showing the cross that generated the double transgenic analyzed here, generated via crossing the Ai14D reporter with a strain bearing the *Slc32a1*^Cre^ transgene (Materials and methods). (**B**) Lightsheet images of the indicated tissues in Ai14D; *Slc32a1*^Cre^ double transgenic animals. (**C**) Lightsheet images of the epididymis in Ai14D; *Slc32a1*^Cre^ double transgenic animals. (**D**) Two representative histology sections from Ai14D; *Slc32a1*^Cre^ cauda epididymis and vas

*Figure 3 continued on next page*

Figure 3 continued

deferens, counterstained with DAPI, clearly showing Tomato-positive principal cells in the epithelium. See also *Figure 3—figure supplement 1*. (**E**) Recombination at the reporter locus in Ai14D; *Slc32a1$^{Cre}$* tissues. Top: Schematic shows location of primers used on the PCR, with expected sizes following recombination of 248 bp (Rec7) and 332 bp (Rec11). Bottom: Each gel shows six tissues from a negative control Ai14D animal (no exposure to Cre), along with six tissues from the Ai14D; *Slc32a1$^{Cre}$* double transgenic. POS: positive control tissue (liver from an Ai14D; *Alb$^{Cre}$* transgenic). Red arrow indicates the band arising from the recombined locus, with clear recombination observed in the brain, as well as all epididymal samples, in Ai14D;*Slc32a1$^{Cre}$* animals.

The online version of this article includes the following figure supplement(s) for figure 3:

**Figure supplement 1.** Flow cytometry analysis of Tomato-positive cells in the epididymis.

section, suggesting either that a subset of principal cells are capable of receiving Cre shipments, or that Cre transfer is extremely inefficient and only a subset of cells are stochastically labeled.

Finally, we sought to determine whether Cre expression in reporter animals resulted in epididymal Tomato signal as a result of Cre itself being shipped from neurons to the epididymis, vs. Cre-induced Tomato produced in neurons being shipped to the epididymis. We therefore assayed recombination at the LoxP-STOP cassette in the epididymis, finding recombination at the reporter locus in genomic DNA isolated from the epididymis (*Figure 3E*). These findings strongly support the hypothesis that Cre activity (RNA or protein) is the relevant molecular signal trafficked from the CNS to the epididymis.

Together, our findings suggest the possibility that Cre RNA or protein synthesized in the CNS is transported to the male reproductive tract, presumably through the circulation (see below). Importantly, the two experimental schemes used to drive Cre expression in the CNS have distinct and unrelated potential artifacts. In the case of AAV transduction, the injected virus could conceivably enter the circulation and might have an unknown tropism for the epididymis. However, this concern does not apply to the use of transgenic constructs to drive Cre expression. Conversely, it is well known that purportedly tissue-specific Cre drivers are often less cell type-specific than expected (*Song and Palmiter, 2018*; *Stifter and Greter, 2020*), and the male reproductive tract – the testis in particular – is well known to express an unusually high fraction of the genome. However, this cannot explain our results using viral transduction to drive Cre expression in the CNS. We further test the hypothesis that Cre is trafficked via the circulation to the male reproductive tract below.

## Many-to-many mapping between Cre drivers and recipient tissues

Our findings thus far suggest the intriguing possibility that the epididymis might be a privileged recipient of neuronally derived molecular cargo. To expand our survey of neuronal Cre lines beyond our initial studies using *Slc32a1$^{Cre}$*, we crossed the Ai14D line to a number of additional CNS Cre drivers: *Fev* (serotonergic neurons), *Gad2* (GABAergic neurons), *Syn1* (pan-neuronal), *Nestin* (pan-neuronal), *Drd1a* (dopaminoceptive neurons), *ChAT* (cholinergic neurons), *Dat1* (dopaminergic neurons), and *Gfap* (glia). We also explored two non-neuronal Cre drivers, crossing the Ai14D animal to *Alb* (liver) and *Adipoq* (adipose tissue) Cre lines. We note that the majority of these genes are either undetectable or expressed at extremely low levels (~1 ppm) in the epididymis as assayed either by bulk or single cell RNA-Seq (*Rinaldi et al., 2020*).

For each of these double transgenic lines, we surveyed a range of tissues, typically including brain, intestine, liver, lung, kidney, testis, seminal vesicle/prostate, and the epididymis. Images for key examples are documented in *Figures 4–5* and *Figure 4—figure supplements 1–4*. Overall, we find that the connection between Cre expression in neurons and reporter expression in the epididymis is not a one-to-one mapping, but rather many-to-many: not only do some non-neuronal Cre lines drive reporter expression in the epididymis, but many Cre lines also drive reporter expression in additional tissues such as the intestine or seminal vesicle. Findings of particular interest are detailed below.

Most notably, we find that a wide range of neuronal Cre lines drive Tomato expression in the epididymis. Focusing first on the putative 'pan-neuronal' Cre lines, driven by the *Synapsin* or *Nestin* promoters, we found extensive Tomato staining throughout the epididymis in both double transgenic lines (*Figure 4—figure supplement 2*). However, we noted that Tomato expression in the *Syn$^{Cre}$* animals was not limited to the epididymal epithelium and instead filled the epididymis lumen (*Figure 4—figure supplement 2A*), suggestive of Tomato-positive sperm. Indeed, offspring of the Ai14D; *Syn$^{Cre}$* double transgenics exhibited Tomato expression throughout the body, consistent with prior reports of *Syn$^{Cre}$* driving germline recombination (*Luo et al., 2020*). Similarly, the *Nestin* Cre

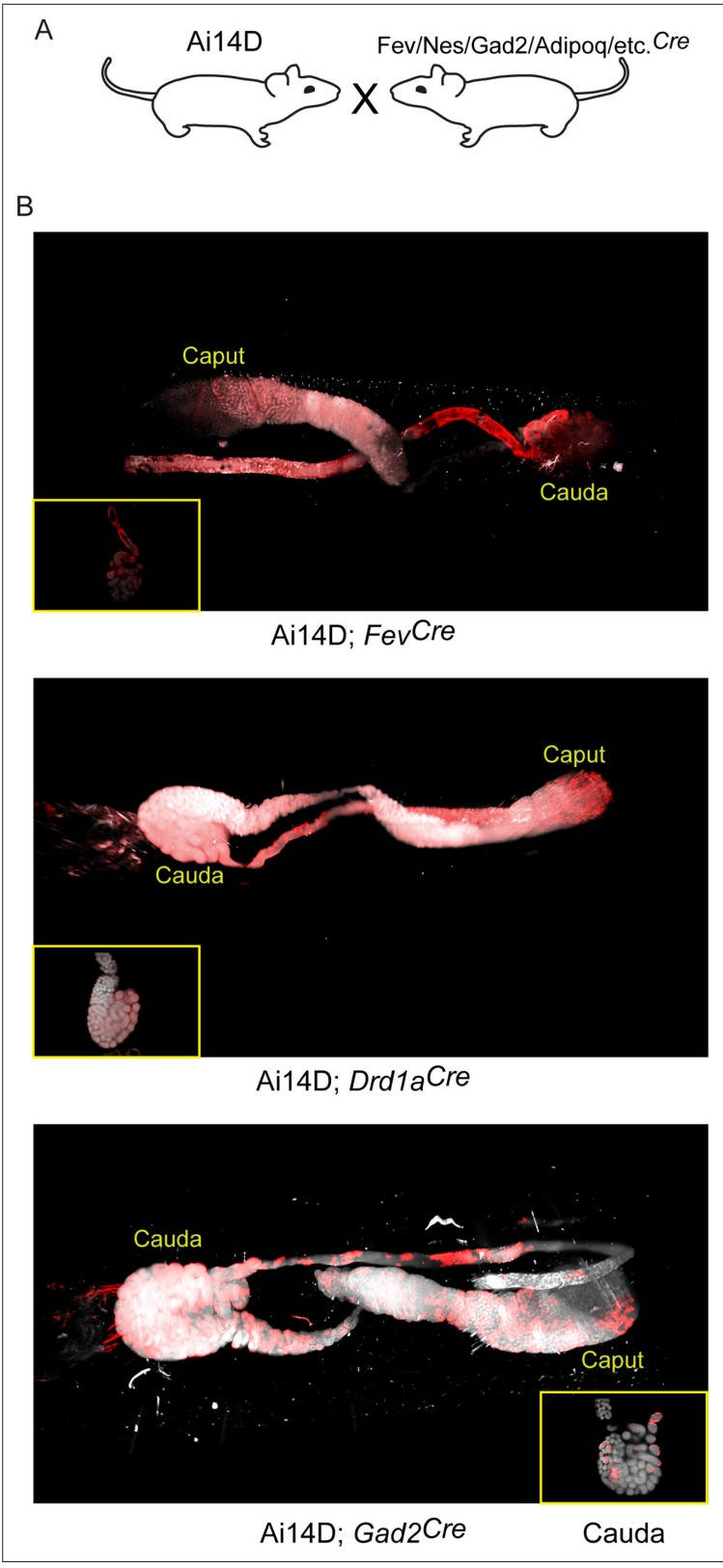

**Figure 4.** Multiple neuronal Cre drivers result in reporter activity in the epididymis. (**A**) Schematic showing the cross that generated the double transgenics analyzed here, as in *Figure 3A*. (**B**) Representative lightsheet images of the epididymis from the indicated double transgenic animals. See also *Figure 4—figure supplements 1–4*.

*Figure 4 continued on next page*

*Figure 4 continued*

The online version of this article includes the following figure supplement(s) for figure 4:

**Figure supplement 1.** Minimal off-target recombination driven by *Alb^Cre*.

**Figure supplement 2.** Tomato expression in epididymis samples from Ai14D animals carrying pan-neuronal Cre drivers.

**Figure supplement 3.** Examples of off-target Cre activity beyond the epididymis.

**Figure supplement 4.** Off-target reporter activity in accessory glands.

driver has also been reported to drive germline recombination (***Betz et al., 1996***; ***McLeod et al., 2020***), and we did obtain Tomato-positive offspring of Ai14D; *Nes^Cre* double transgenics. That said, in contrast to the Ai14D; *Syn^Cre* animals we found no detectable labeling of sperm in Ai14D; *Nes^Cre* animals, with Tomato expression in the epididymis largely confined to the epididymal epithelium (***Figure 4—figure supplement 2B***). This suggests that *Nes^Cre*-mediated recombination occurs relatively late during spermatogenesis in germ cells, with recombination in the epididymal epithelium occurring independently – probably induced by the same CNS→epididymis trafficking seen for several other neuronal Cre drivers (***Figure 3*** and below).

Turning from pan-neuronal promoters to more cell type-specific CNS Cre drivers, we consistently observed Tomato expression in the epididymis when Cre was expressed from *Slc32a1*, *Fev*, *Gad2*, and *Drd1a* promoters (***Figures 3–4***). For two of these Cre drivers – *Fev* and *Slc32a1* – we find that Tomato expression was strongest in the cauda epididymis and vas deferens, similar to our findings obtained using AAV to drive Cre expression in the CNS. In contrast, both Ai14D; *Gad2^Cre* and Ai14D; *Drd1a^Cre* double transgenic animals exhibited patchy epithelial staining throughout the epididymis (***Figure 4***). Intriguingly, off-target epididymal recombination was not unique to neuronal Cre drivers, as we also observed robust Tomato expression and reporter locus recombination in animals carrying the adipose-specific *Adipoq^Cre* (***Figure 5***). In contrast, reporter animals carrying the liver-specific *Alb^Cre* transgene exhibited barely detectable Tomato expression in the epididymis with few Tomato-positive cells detected throughout the entire tissue – at most one or two cells per longitudinal section (***Figure 4—figure supplement 1***) – indicating that not all Cre transgenes drive extensive recombination in the epididymis.

Thus, although our initial studies suggested a privileged molecular trafficking pathway from the CNS to the male reproductive tract (***Figures 2–4***), the robust Tomato expression driven by the adipose-specific *Adipoq* Cre driver (***Figure 5***) suggested the possibility of additional avenues for unanticipated inter-tissue Cre trafficking. Not only did non-neuronal Cre drivers direct recombination in the epididymis, but multiple neuronal Cre drivers exhibited off-target activities in tissues beyond the epididymis. Most notably, within the male reproductive tract we observed robust Tomato expression in the reproductive accessory glands of the *Nes*, *Fev*, and *Gad2* lines (***Figure 4—figure supplements 3–4***).

## Off-target recombination in the epididymis is not idiosyncratic to the Ai14D model

Our data demonstrates that a wide range of putatively tissue-specific Cre lines drive recombination in the epididymis of Ai14D animals. Given the extraordinary nature of this finding, we were concerned that our results might reflect some idiosyncracy of the LoxP-flanked Tomato reporter locus in the Ai14D strain. We first explored publicly available annotations for Cre drivers from MGI (Mouse Genome Informatics: https://www.informatics.jax.org, data retrieved repeatedly from ~2018 through October 2022), which includes images from a variety of Cre drivers crossed to a LacZ reporter line. Of the lines that drive recombination in Ai14D in our study, there is also clear evidence for recombination in the epididymis epithelium for *Syn1^Cre*, along with 'ambiguous' expression annotated for *Nes^Cre* and *Gad2^Cre*. However, closer examination of the image provided for *Nes^Cre* also revealed clear evidence for reporter expression in the epididymis epithelium; no image was available for *Gad2^Cre*. These reporter assays thus are consistent with our findings using the Ai14D reporter animal. Conversely, *Adipoq*, *Slc32a1*, *Drd1*, and *Fev* were all annotated as 'absent' in the epididymis epithelium (at postnatal days 7 and 56). However, given that epididymal activity was annotated as 'ambiguous' for *Nes^Cre* despite clear microscopic evidence for reporter expression, we were concerned about false negative annotations for these other Cre drivers, for which either no images were available for scrutiny, or

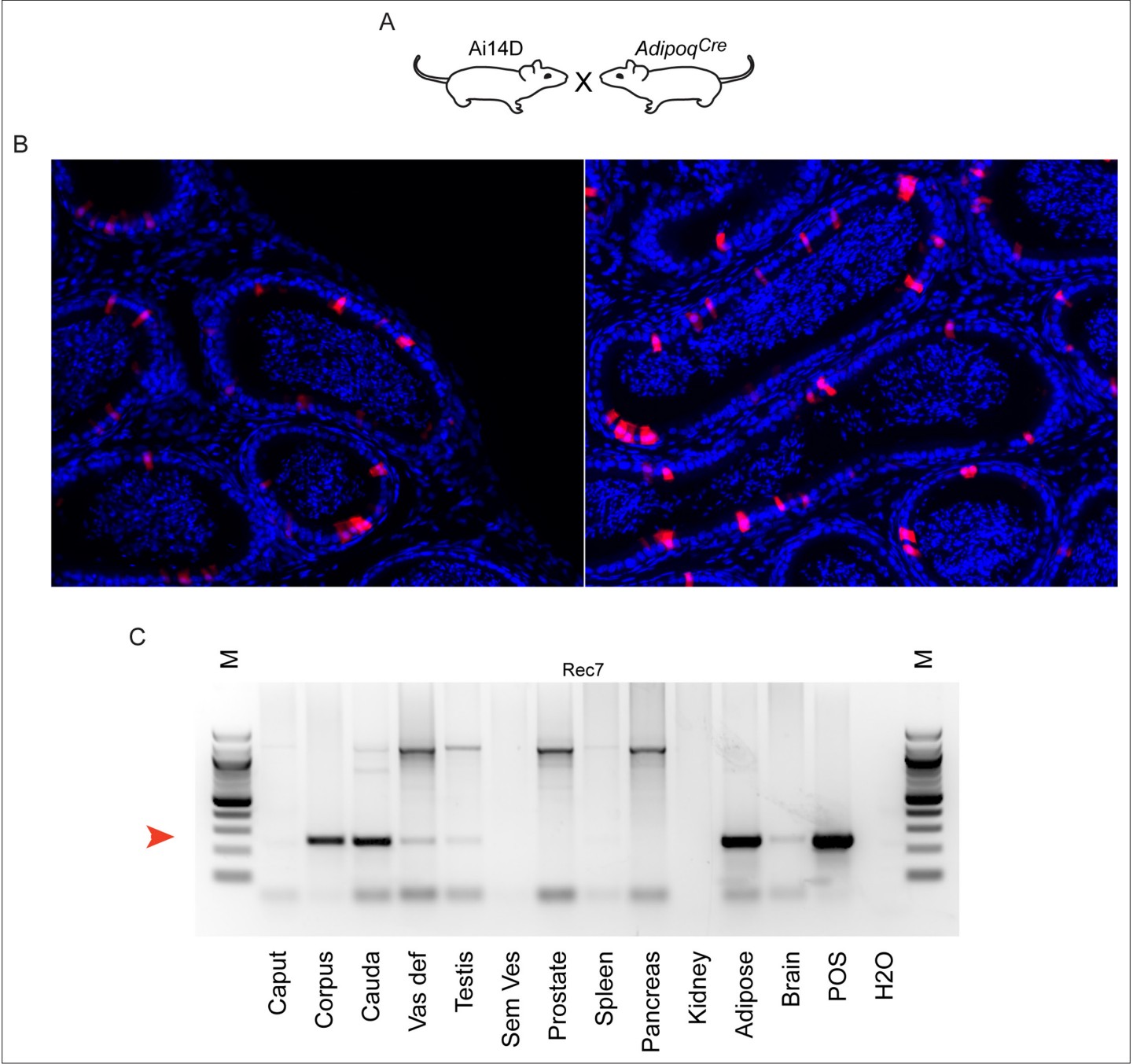

**Figure 5.** Reporter activity in *Adipoq^Cre* animals. (**A**) Schematic, as in *Figures 3A and 4A*. (**B**) Histology from Ai14D; *Adipoq^Cre* animals, showing Tomato expression scattered throughout the epididymis epithelium. (**C**) PCR analysis of indicated tissues of Ai14D; *Adipoq^Cre* animals, revealing robust recombination in the epididymis. POS: positive control tissue. Red arrow indicates the band arising from the recombined locus. See also *Figure 5—figure supplement 1*.

The online version of this article includes the following figure supplement(s) for figure 5:

**Figure supplement 1.** *Adipoq* Cre drives recombination of a second target locus in the epididymis.

the available image was limited to a very narrow region of the epididymis. Nonetheless, the lack of annotated recombination in this dataset, for Cre drivers which exhibit robust activity in our system, suggested the possibility that some reporter loci could be relatively resistant to Cre activity – by virtue of the target locus being packaged in relatively inaccessible chromatin in the epididymis, for instance – compared to the Tomato reporter locus present in the Ai14D animal.

To extend our findings to other target loci, we therefore obtained double transgenic animals carrying the *Adipoq* Cre driver along with a conditional allele of *Dicer1* with LoxP sites flanking exons 21 and 22 (*Figure 5—figure supplement 1A*). PCR genotyping confirmed the expected robust recombination in adipose tissue, while tail tip DNA was used as a negative control (*Figure 5—figure supplement 1B*). Using this assay, we document recombination of this locus specifically in the cauda epididymis, but not in the testis or other parts of the epididymis (*Figure 5—figure supplement 1B*). These data thus extend our results to a second target locus, demonstrating that off-target recombination in the epididymis is not unique to the Ai14D reporter.

## Cre activity likely traffics through the circulation to target the epididymis

Together, our findings reveal that the epididymis is a surprisingly common target of a variety of Cre drivers that are not generally thought to be expressed in the male reproductive tract. While the observations made using double transgenic lines could result from leaky expression of the promoters in question – whether in adulthood or during early development of the epididymis – this hypothesis would not explain the results observed following AAV-mediated Cre expression in the brain (*Figure 2*). Nonetheless, we sought to definitively test the hypothesis that Cre activity, whether protein or RNA, synthesized in distant tissues can make its way to the male reproductive tract through the circulation.

As an initial test of this hypothesis, we generated parabiotic animal pairs (*Gibney et al., 2012*), in which two animals are surgically joined (hereafter represented as animal1::animal2) to establish a conjoined circulatory system (*Figure 6A*, *Figure 6—figure supplement 1A–C*). Here, parabioses were established between a Cre driver line (focusing on the *Slc32a1* and *Nestin* Cre lines) and an Ai14D reporter animal. Eight to nine weeks after establishing the parabiosis, we sacrificed the animal pairs and harvested tissues for analysis. We detected moderate Tomato expression in the epididymis of the Ai14D recipient animal in a parabiotic Ai14D:: *Slc32a1^{Cre}* pair (*Figure 6B*, *Figure 6—figure supplement 1B*), along with a faint band supporting recombination at the reporter locus in one recipient (*Figure 6C*). However, in two other Ai14D animals we found no evidence for recombination by PCR (*Figure 6—figure supplement 1C*), motivating more extensive characterization of the potential for Cre transfer through the circulation.

Given the technical and logistical challenges involved in generating large numbers of parabiotic animal pairs, we attempted to transfer any potential circulating Cre activity by injecting serum or circulating exosomes/extracellular vesicles (EVs) (enriched via tangential flow filtration [TFF] *Heinemann et al., 2014*), from various Cre lines into Ai14D reporter animals (*Figure 6D*). In an initial experiment, we obtained serum from a *Slc32a1^{Cre}* animal and transferred it into the tail vein of an Ai14D male. This procedure was repeated every other day, for a total of three transfers over 5 days. As shown in *Figure 6E–F*, this resulted in epididymal LoxP recombination and Tomato expression in animals receiving *Slc32a1^{Cre}* serum, while recipients of serum from FVB or *Alb^{Cre}* animals did not express Tomato or exhibit recombination at the reporter locus.

Follow-up studies were, unfortunately, somewhat ambiguous (*Figure 6—figure supplement 1D–F*). We initially set out to leverage the serum transfer approach to fractionate serum and identify the circulating material responsible for driving recombination in recipient animals. We found no Tomato expression in an Ai14D male that had received TFF-enriched exosomes from nine *Slc32a1^{Cre}* animals spread across 3 injection days (*Figure 6—figure supplement 1E*, top right panel). Across many subsequent experiments, we used a range of donor animals carrying various Cre drivers and varied our injection scheme to focus on whole serum, concentrated exosomes, or combinations of the two. Ultimately, across over 20 recipient animals we documented clear evidence for recombination in six animals, with positive results in recipients of *Slc32a1^{Cre}* serum, *Slc32a1^{Cre}* serum and exosomes, *Gad2^{Cre}* serum and exosomes, a mixture of *Gad2^{Cre}* and *Drd1^{Cre}* serum and exosomes, *Nes^{Cre}* serum and exosomes, and *Adipoq^{Cre}* serum and exosomes (*Figure 6—figure supplement 1E–F*). However, similar or nearly identical injection schemes often resulted in no detectable Tomato expression or recombination. Although we attempted to account for variables ranging from the recipient animal's age, to injection time during the day, we have not been able to identify variables that reliably distinguish injection schema that lead to recombination from those that fail.

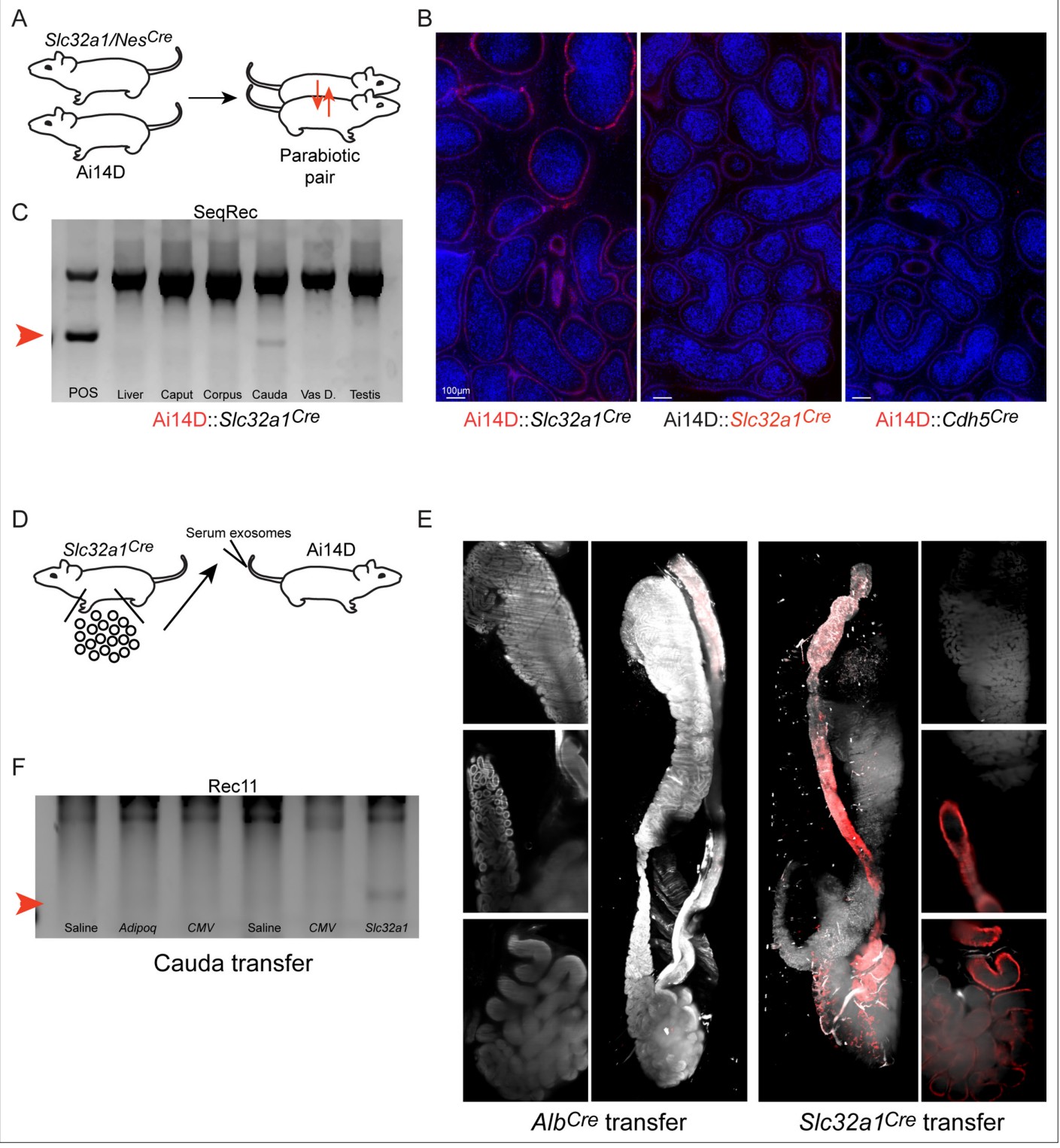

**Figure 6.** Cre activity present in the circulation can be transferred from a Cre driver to a reporter animal. (**A**) Experimental schematic. Parabiosis surgery was used to link the circulatory systems of an Ai14D reporter animal and that of a *Slc32a1^Cre* or *Nes^Cre* animal. After 8–9 weeks together, animals were sacrificed and tissues from both animals were obtained for analysis. (**B**) Histology from the indicated sides (highlighted in red) of two Ai14D::Cre parabiotic pairs. We observe Tomato expression specifically on the reporter side, but not the Cre side, of Ai14D::*Slc32a1^Cre* animals. (**C**) Genotyping PCR from the Ai14D side, and the Cre side, of the indicated parabiotic pairs, with faint recombination detected in the cauda epididymis of Ai14D::*Slc32a1^Cre* animals. (**D**) Schematic showing serum/exosome transfer experiments. Note that these experiments were carried out with a wide range of injection schedules (Materials and methods), using both whole serum transfers and TFF-enriched exosome transfers for recipient animals. (**E**) Lightsheet images

*Figure 6 continued on next page*

*Figure 6 continued*

of Ai14D recipient following transfer of serum and exosomes from *Alb^Cre* donors (left panels), or from *Slc32a1^Cre* donors (right panels). (**F**) Genotyping PCR from cauda epididymis samples obtained from Ai14D animals receiving serum or TFFs from various Cre donor lines. Red arrow indicates the band diagnostic of a recombined locus.

The online version of this article includes the following figure supplement(s) for figure 6:

**Figure supplement 1.** Inconsistent Cre transfer via parabiosis or serum transfer experiments.

**Figure supplement 2.** Cre-encoding mRNA present in the circulation of various Cre driver animals.

Taken together, our data support the hypothesis that off-target Cre activity in the epididymis is likely mediated by Cre trafficking through the circulation, although the difficulty in reliably replicating this transfer impedes further mechanistic follow-up.

## Discussion

Our observations reveal the epididymis as a surprisingly common 'off-target' tissue for a variety of Cre drivers that are not believed to be expressed in the male reproductive tract. We find a many-to-many mapping, with 'sender' tissues including multiple neuronal subpopulations along with adipose tissue, and recipient tissues including the epididymis, seminal vesicle, prostate, and intestine. Importantly, different Cre lines drive recombination in different sets of tissues, arguing against the possibility that a generic 'sink' tissue nonspecifically receives any RNAs or proteins present in the circulation.

These findings are most generally of interest to the mouse genetics community. It has long been clear that transgenes carrying Cre under the control of a putatively cell type-specific promoter are often expressed in unexpected tissues (*Song and Palmiter, 2018*; *Stifter and Greter, 2020*), and here for example our data confirm prior reports of germline activity of *Nes^Cre* and *Syn1^Cre* drivers (*Luo et al., 2020*; *Betz et al., 1996*; *McLeod et al., 2020*). Our findings also identify many new examples of off-target Cre activity for a variety of Cre lines, many of which may have been missed before given the relatively small size of the scientific community focused on the epididymis. In this context our data simply further emphasize the importance of characterizing recombination in conditional knockout studies, as recombination in unanticipated tissues has the potential to confound interpretations of tissue-specific functions of genes of interest. Given the common off-target recombination in the epididymis, seminal vesicle and prostate documented here, this is most obviously true for any studies focused on male reproductive biology, but as we also find unexpected Cre activity in intestine driven by neuronal Cre drivers we emphasize more general caution when using Cre-Lox genetics in mouse.

The impact of these off-target effects will of course depend on the nature of the conditional allele used in a given study. It is worth noting that even in the cases of double transgenics exhibiting robust recombination in the epididymis (*Slc32a1*, *Adipoq*, etc. – *Figures 3–5*), we find fairly patchy recombination in the epididymal epithelium. This mosaic recombination behavior has different implications for conditional knockouts, where a large number of epididymis (or seminal vesicle, etc.) cells will escape deletion, vs. gain of function studies where even patchy Cre-dependent induction of gene expression (such as the tdTomato reporter used here) in the target tissue has the potential to exhibit artifactual gain of function.

### What is the mechanistic basis for off-target activity of neuronal Cre drivers?

The fact that putatively tissue-specific Cre drivers can often drive recombination in unanticipated tissues has long been known. This is commonly thought to reflect 'leaky' expression of a given Cre transgene, potentially owing to the removal of a given promoter from its genomic context (and thus from long-range regulatory elements) in many Cre drivers. In addition, while many promoters chosen as cell type-specific Cre drivers are highly expressed in the cell type of interest, they are often also expressed at lower levels in other cell types throughout the body where the Cre transgene therefore has the potential to drive recombination.

That said, our data showing off-target Cre activity in the epididymis are unlikely to result from leaky expression of the Cre drivers studied here, for multiple reasons. First of all, we find a wide range of Cre transgenes drive recombination in the epididymis, and most of the neuron-specific genes in

question are not detectably expressed in the adult epididymis (*Johnston et al., 2005*; *Rinaldi et al., 2020*). It remains plausible that these genes could be expressed in the epididymis only during early development, before becoming undetectable in the adult epididymis. However, our observation that induction of Cre expression in the brain via AAV-mediated transduction leads to reporter activity in the epididymis epithelium (*Figure 2*, *Figure 2—figure supplements 1–2*) cannot be explained by leaky transgene expression. Conversely, the possibility that unanticipated tropism of the AAV constructs used here might result in viral transduction of the epididymis would not explain any of the results obtained using double transgenics (*Figures 3–5*). Together, these findings strongly suggest that Cre activity synthesized in neurons is transported to the epididymis.

The delivery of Cre from neurons to the epididymis could occur via a number of routes, including release of Cre-containing vesicles by peripheral neurons that innervate the cauda epididymis. We consider this to be unlikely for several reasons. First, we show that adipose-specific Cre expression reliably drives recombination in the epididymis (*Figure 5*), arguing against Cre transfer requiring neuronal termini. Second, induction of Cre expression specifically in one hemisphere of the brain via stereotactic AAV injections was shown to lead to Tomato expression in both the ipsilateral and contralateral epididymis (*Figure 2*), arguing against short-range neuronal delivery (not to mention the challenges in considering how Cre might move from the CNS to peripheral nerves).

Indeed, our follow-up studies using parabioses and serum transfers support the hypothesis that Cre activity synthesized in a variety of tissues can access the circulation and subsequently be trafficked to the male reproductive tract. Although these efforts to transfer Cre activity from a Cre-expressing animal to a reporter animal were inefficient, the fact that recombination was *ever* observed in these studies is best explained by the trafficking hypothesis. We speculate that the circulating Cre activity consists of *Cre* mRNA, rather than protein, and we anticipate that the RNA is present either in exosomes or in ribonucleoprotein complexes protected from serum nucleases. Indeed, we can document *Cre* mRNA by PCR in the circulation of multiple Cre lines (*Figure 6—figure supplement 2*), and there is abundant precedent for RNA transmission between different tissues. We cannot presently determine whether the transferred Cre activity is contained in exosomes, microvesicles, ribonucleoprotein complexes, or other forms. Again, precedent argues for exosomes as the carriers of inter-tissue molecular information (*Valadi et al., 2007*), but given the inefficiency of Cre transfer in serum and exosome injections (*Figure 6*, *Figure 6—figure supplement 1*), a detailed fractionation effort would require an inordinate number of animals to be used, and must be reserved for future studies.

## Potential biological implications

Discovery of a pathway by which neurons (and some other tissues) potentially traffic RNAs (or proteins) to the male reproductive tract would have exciting implications. Most intriguingly, we speculate that environmental conditions experienced by a male could be interpreted by the CNS, with salient features of the environment then communicated to the male reproductive tract via this trafficking system. Given the emerging role for the epididymis in remodeling multiple aspects of maturing sperm (*Gervasi and Visconti, 2017*; *Tamessar et al., 2021*; *Zhou et al., 2018*), an appealing hypothesis would be that receipt of CNS-derived information – whether RNAs (mRNAs, lincRNAs, sncRNAs, etc.), proteins, etc. – could then influence epididymal physiology and thus alter the post-testicular maturation of germ cells in the male reproductive tract.

Interestingly, at least two prior studies in the paternal effects literature have suggested the possibility that circulating information may play a role in reprogramming the sperm epigenome. First, in their studies of intergenerational effects of carbon tetrachloride, Zeybel et al. reported that repeated serum transfers from $CCl_4$-treated animals to a recipient were sufficient to reproduce H3K27me3 changes in the sperm of the recipient animal (*Zeybel et al., 2012*). More recently, Mansuy and colleagues reported that transferring serum from males subject to early life stress to a recipient male was sufficient to drive altered glucose responses in the F1 offspring of the recipient animal (*van Steenwyk et al., 2020*), again suggesting that circulating factors may play a role in reprogramming the sperm epigenome. Although neither of these studies implicated circulating RNAs in modulating sperm biology – Zeybel et al. did not speculate about the nature of the soluble factor(s), while van Steenwyk et al. focused on lipids with the potential to activate various nuclear hormone receptors – they highlight the potential for circulating factors to mediate communication between distant tissues and the germline.

This possibility of inter-tissue RNA shipments between distant tissues and the germline is particularly intriguing considering the role of systemic RNA trafficking in worms and plants. Most relevant to our study, small RNAs produced in *C. elegans* neurons have been documented to silence genes in the germline and induce transgenerational silencing (*Devanapally et al., 2015*). Intriguingly, a prior report in mouse suggested that microRNAs expressed in the brain could be delivered to sperm, and subsequently to the next generation (*O'Brien et al., 2020*). However, with the exception of *Syn1* and *Nestin* (*Figure 4—figure supplement 2*), we do not find any evidence for neuronal Cre trafficking directly to sperm cells, arguing against the idea that the circulating Cre activity is stably delivered to sperm. Even in the case of *Syn1* and *Nestin* Cre drivers, recombination in the germline likely occurs prior to the last stages of spermatogenesis in the testis – potentially in spermatogonial stem cells or even earlier in development – and not during epididymal transit. Our data instead would place the epididymis as a mediator between neuronal information processing and programming of the sperm epigenome, with CNS-derived information potentially driving altered epididymal physiology and consequent remodeling of sperm small RNAs, proteins, or surface lipids.

An alternative explanation of our findings would be that neuronally derived molecular information does not have any physiological consequences upon uptake by the epididymis. In such a 'trash can' model, the epididymis could simply be unusually receptive to circulating exosomes or RNPs, with the unique sensitivity of Cre-Lox systems revealing the delivery of Cre molecules at abundances that would likely be meaningless for a typical cellular sncRNA, mRNA, or protein. We disfavor this model, in part because different Cre drivers result in recombination in different parts of the epididymis, arguing for some level of specificity between different tissue-derived exosomes (for instance) and target cell populations in the epididymis.

Nonetheless, without any evidence for physiological functions for this trafficking system we cannot formally rule out the trash can hypothesis at present; we envision two potential approaches to this question in the future. One possibility would be to subject male mice to some stress paradigm or other exposure, then identify changes in transcription or small RNA production in the epididymis. Subsequently, males could be subject to the exposure in question, with circulating serum/exosomes then isolated and injected into an unexposed recipient. In this scenario, if circulating material is responsible for programming epididymal gene expression, the recipient animal would be expected to exhibit the molecular changes seen in directly exposed animals. While this approach could be very powerful, our experience with serum/exosome transfers has been that exosome transfer results in very inefficient transfer of Cre activity between animals, with exosomes from at least nine donor animals required for patchy and low-penetrance reporter activity (*Figure 6*, *Figure 6—figure supplement 1*). This suggests that serum from scores of animals might be required for a single recipient to recapitulate the epididymal changes driven by direct exposures.

An alternative approach would be to identify the molecular basis for epididymal receptivity to neuronal information. For instance, if the circulating carrier of neuronal information is indeed present in exosomes, tetraspanins or other cell surface molecules are presumably responsible for tissue-specific targeting of exosomes to the epididymis. Identification of the relevant exosome receptor could then be used in knockout studies to prevent receipt of neuronal shipments, and changes in epididymal responses to various organismal stressors in wild type vs. knockout animals could then be used to identify potential effects of neuronal RNA/protein shipments on epididymal physiology.

## Conclusions

Here, we have demonstrated that Cre activity synthesized in neurons and in adipose tissue consistently directs recombination in the epididymis – with some lines also having other off-target tissues. These findings add to the literature on the dangers of Cre-Lox genetics in conditional knockout studies, identifying additional off-target tissues for a number of Cre drivers. In addition, our data suggest that this poor tissue specificity could be caused in part by mobilization of Cre activity through the circulation, implying a surprising mechanism behind some of these off-target effects. Finally, the possibility that this pathway plays a role in sensing of environmental conditions in modulation of the sperm epigenome represents an exciting hypothesis for future studies in paternal effect epigenetics.

# Materials and methods

## Mouse husbandry

All animal use was approved by the Institutional Animal Care and Use Committees of UMass Chan Medical School and Harvard Faculty of Arts and Sciences, under protocol PROTO202100029 to Dr Oliver Rando and protocol 29014 awarded to Dr Amy Wagers. Double transgenic *Adipoq*^Cre; *Dicer1*^flx/flx animals were a gift from Bruna Brandao and C Ronald Kahn.

The transgenic mice utilized in this study were maintained by filial mating, giving preference toward homozygous carriers whenever possible. To avoid introduction of genetic confounders the breeding colonies were maintained for up to three generations, after which animals were either backcrossed to their background strain or the colony was culled, and new breeding pairs acquired.

Transgenic animals were typically sacrificed for imaging at 10–12 weeks of age, unless otherwise noted. AAV injections were carried out between 8 and 12 weeks of age. For parabiosis experiments, animals underwent surgery to establish conjoined circulation between 21 and 28 days after birth. Finally, for serum/exosome transfer experiments, recipient animals were 10–15 weeks, while donors of plasma were typically older than 15 weeks of age as this afforded more material for injections.

Below is a list with the official names for all the mice strains and the genotype abbreviations used throughout this text, as well as the strain number listed at the Jackson Laboratory.

| Transgenic allele | Expression of | Promoter | JAX catalog # | Genotype abbreviation |
|---|---|---|---|---|
| B6.Cg-Gt(ROSA)26Sortm14(CAG-tdTomato)Hze/J | TdTomato upon Cre | Chicken Actin Gene | 007914 | Ai14D |
| Slc32a1tm2(cre)Lowl/JW | Cre | Vgat – GABA vesicular transporter | 028892 | *Slc32a1*^Cre |
| B6.Cg-Tg(Nes-cre)1Kln/J | Cre | Nestin | 3771 | *Nes*^Cre |
| B6N.Cg-Speer6-ps1Tg(Alb-cre)21Mgn/JR | Cre | Albumin | 003574 | *Alb*^Cre |
| B6.FVB-Tg(Adipoq-cre)1Evdr/J | Cre | Adiponectin | 028020 | *Adipoq*^Cre |
| B6;129-Tg(Drd1a-cre)120Mxu/MmjaxT | Cre | Drd1a – dopamine receptor G-protein | 037156 | *Drd1*^Cre |
| B6N.Cg-Gad2tm2(cre)Zjh/JT | Cre | Gad2 – glutamic acid decarboxylase 2 | 010802 | *Gad2*^Cre |
| B6.Cg-Tg(Fev-cre)1Esd/JT | Cre | Fev – ETS oncogene family | 012712 | *Fev*^Cre |
| B6.Cg-Tg(Syn1-cre)671Jxm/J | Cre | Synapsin1 | 003966 | *Syn1*^Cre |
| STOCK Defb41tm1(icre)Psip/R | iCre | Defb41 – Defensin beta 41 | Gift from P Sipila | *Defb41*^iCre |
| B6.C-Tg(CMV-cre)1Cgn/J | Cre | CMV | 006054 | *CMV*^Cre |

All genotypes were determined using Transnetyx genotyping services.

## AAV constructs

An initial set of three animals were injected with rAAV2-retro-Eif1a-Cre. Subsequent experiments used an AAV9-based construct carrying pCMV-CRE: AAV9.CB-PI-Cre 5.03×10^12 GC/ml; VCAV-04562. Virus was a gift from the Gao lab.

## Stereotaxic AAV injections

Stereotaxic injections were performed to deliver AAV constructs into different regions of the CNS of adult Ai14D males. Adult male mice (8–12 weeks of age) were anesthetized with a mixture of 100 mg/kg ketamine and 10 mg/kg xylazine, administered via intraperitoneal injection. Prior to surgery, the top of the skull was shaved and disinfected. Surgeries were performed using aseptic technique with

the aid of a stereotaxic frame (Stoelting Co.). Mice were placed into the stereotaxic frame and the skull was exposed by making a small incision with a scalpel blade. Using bregma and lambda as landmarks, the skull was then leveled along the coronal and sagittal planes. Two small drill holes were made in the skull that allowed injections to target the somatosensory cortex, caudate putamen, and nucleus accumbens. Microinjections were made using a Hamilton 10 µl neurosyringe (1701RN; Hamilton) and a microsyringe pump (Stoelting Co); 0.5 µl of virus was delivered through the syringe and at a constant flow rate of 30 nl/min. After injection, the needle was left unmoved for 10 min before being slowly retracted. The incision was then closed and held together with glue.

## Tissue collection and preparation

### Transcardiac perfusion

To prepare tissues for imaging, animals were subject to transcardiac perfusion under deep anesthesia. Once deep anesthesia, assessed by toe pinch, was established, the thoracic cavity was opened to expose the heart and heparin was directly infused into the left ventricle using an insulin syringe with 31-gauge needle. A butterfly needle hooked to a peristaltic pump was secured in the left ventricle and the right atrium nicked to allow perfusion with 20 ml PBS, followed by 25 ml of freshly prepared 4% paraformaldehyde (PFA). Harvested organs were incubated in 4% PFA overnight, then washed with PBS and either transferred to 30% sucrose for cryoprotection (followed by OCT embedding for histology) or subjected to tissue clearing for lightsheet imaging.

For tissue clearing, passive de-lipidation performed by incubating the organ with 4% SDS/PBS at room temperature (RT) in an orbital shaker for 3 weeks (*Rinaldi et al., 2018*). The 4% SDS/PBS solution was changed every other day. At the end of the third week, the organ was rinsed with PBS for three times of least 4 hr and finally placed on a refractive index matching solution (0.17 M iodixanol; 0.4 M diatrizoic acid, 1 M *n*-methyl-d-glucamine, 0.01% sodium azide) 24 hr prior to imaging. Light-sheet images were acquired using LaVisionBioTec LightSheet Microscopy Ultramicroscope II located at the Cornell Imaging facility, Ithaca, NY. Images were processed using Arivis 4D software.

## Parabioses

Male mice of similar age from two different strains – one 'donor' animal expressing *Cre*, and the Ai14D *Tom*^flx/flx^ 'recipient' – were surgically connected to allow the sharing of their circulatory systems. Surgical preparations were performed on the shaved skin, where incision was made along the opposing flanks of each mouse to be joined. A skin flap was made without damaging the underlying peritoneal lining, and the mice placed side-by-side in a prone position. To better secure the connection, the animals' joints (elbows and knees) were held together using suture. The dorsal and ventral skin flaps from the different mice were then pinched together and sutured. Parabiotic pairs were maintained for 8 weeks after surgery and therefore expected to have maintained cross-circulation for at least 2 weeks (*Wright et al., 2001*).

## Serum and exosome injections

### Serum collection

Blood was collected from the right ventricle of the heart of a deeply anesthetized animals using a 3 ml syringe 27-gauge needle. If serum was to be used for TFF, ultracentrifugation, or RNA extraction, a more thorough exsanguination was achieved by injection of 1.5 ml of saline solution using a 31-gauge needle into left ventricle, thus pushing more blood to the right ventricle for collection.

The blood was transferred to the yellow cap serum collection tubes and placed on ice until the end of the collection. Once all animals were processed, serum was obtained after incubating samples for 20 min at RT and performing a 9000 × *g* spin for 10 min followed by 12,000 × *g* for 5 min. Serum was transferred to a clean 1.5 ml Eppendorf tube further centrifuged at 4°C for 10 min at 16,000 × *g*, and serum transferred to a new tube and either used for intravenous injections or stored at –80°C for later processing.

### TFF exosome enrichment

We use the manual TFF-easy from Galen molecular laboratory supplies. The device is a filter cartridge with polysulfone hollow fibers with pores of 5 nm diameter along its length. It allows the concentration of EVs by the removal of small proteins, molecules, and solutes from serum. Water and small molecules

up to 5 nm pass through the pores and this fluid is discarded as flowthrough thus allowing the concentration of the serum EVs. https://galenmolecular.com/exosome-products/exosome-isolation/tff-easy/.

### Injection schedules

Serum and exosome injections were performed using serum/exosomes isolated from a variety of Cre lines, with at least three tail vein injections (~250 µl per bolus) per recipient spread across 5 days (e.g. every other day). Across all injection schemes, epididymal recombination was observed only in a subset of recipients (*Figure 6* and *Figure 6—figure supplement 1*). We therefore continuously modified the injection scheme used – hoping to identify reproducible conditions for transfer of Cre activity – with various recipients receiving serum only, EVs only, or various mixtures of the two. Whenever possible, experiments were performed using littermates with two animals receiving serum/EV and two receiving saline. Treated and untreated animals were kept in different cages.

The transfer experiments using the maximal amount of material involved transfer of TFF-enriched EVs obtained from plasma pooled from 10 to 12 animals on days 1, 5, and 10, as well as serum pools from two to three mice injected on days 3, 8, and 12. Recipient animals were sacrificed for analysis on day 15. Unfortunately, across all trials we have been unable to identify experimental variables that distinguish successful (as defined by reporter expression in the recipient animal) from unsuccessful transfers.

## Histology, tissue clearing, and lightsheet microscopy

### Histology

Male mice from the same strain and age as previously described were perfused with 4% PFA/PBS, and epididymi explanted and further incubated in fixative at 4°C overnight (ON). After washing the excess of PFA with PBS, the sample was incubated at 4°C with a 30% sucrose, 0.002% sodium azide in PBS solution for 32 hr, or until tissues sank to the bottom of the container. Prior to mounting in OCT, sucrose was replaced with a one-to-one solution of 30% sucrose to OCT and samples kept at 4°C with agitation. Samples were than mounted using OCT. Cryo-sections were done at 5–10 µm thickness by the UMASS morphology core. Slides were stored at –80°C until further processing.

### Fluorescence

Frozen slides were incubated at a 37°C warm plate for at least 10 min to ensure proper attachment of the section to the slide, then washed 3 × 5 minutes in PBS 0.02% Tween 20 (PBS-T) to remove OCT. For the nuclear visualization, slides were immersed in a PBS solution containing 3 µM DAPI (4',6-diamidino-2-phenylindole in PBS) for 5 min. Afterward slides were rinsed and mounted with ProLong Antifade gold (Thermo Fisher) and imaged on an Automated Zeiss Axioobserver workstation with DIC, LED fluorescence, autofocus, and deconvolution image processing.

## Flow

Single cell suspension of the epididymis were performed as previously described (*Rinaldi et al., 2020*). Briefly, epididymis samples were separated into caput corpus cauda and vas and enzymatically dissociated. Cells were resuspended in HSS and transferred to a 1.5 ml Eppendorf tube for staining with viability dye and nuclear stain. Afterward the cell suspension was once more filtered while being transferred to a 5 ml polypropylene FACS tube, placed in ice, and protected from light until flow cytometry data acquisition. Data acquisition was performed in Cytec or LSRII with PMT voltages adjusted for each experimental day using single stained cells from homozygous Ai14D mice as well as from ubiquitously Cre-expressing Ai14D mice. Fluorescence minus one was also made. No compensation was necessary.

## PCR assay for recombination

Recombination PCR was done using 2× MyTaq Red Mix (Meridian Biosciences, cat. BIO-25044) per manufacturer's instructions. Primers (*Supplementary file 1*) were diluted to final concentrations of 0.33 µM for Rec7, 0.67 µM for Rec11 and SeqRec, and 0.5 µM for *Dcr1* KO. Two-hundred ng DNA was used per reaction. PCR products were run on 2% agarose gel.

### High throughput RNA/DNA extraction

Slightly modified from the described steps on the Qiagen RNA&DNA extraction protocol. Namely, mandatory use of the antifoam reagent and adoption of a fixed 1 ml volume of lysis buffer to 3 mm$^3$ of tissue. The best lysis was obtained using bead beater with the pre-programmed settings for kidney mouse tissue repeated three times and starting every cycle on frozen tissue/lysis buffer.

## Acknowledgements

We thank Bruna Brandao and C Ronald Kahn for the generous gift of *Adipoq*Cre; *Dicer1*flx/flx animals, Guanping Gao for the generous gift of AAV9-CMV-Cre, Andrew Recknagel for tissue and images, and Dr Rebecca Williams and grant S10 OD023466 – A Light Sheet Microscope for the Cornell BRC Imaging Facility – for lightsheet microscopy help. We thank I Bach, R Davis, and D Weaver for generous gift of several Cre driver lines, and D Weaver and P Visconti for critical reading of the manuscript. This work was funded by the Templeton Foundation (Grant 61350) and NICHD (R01HD080224) to OR, NIH (R01DA047678) to ART, and NIH (DP1 AG063419) to AJW.

## Additional information

### Competing interests

Amy J Wagers: is a scientific advisor for Kate Therapeutics and Frequency Therapeutics and a cofounder, advisor, and holder of private equity in Elevian, Inc, which also provides sponsored research to the Wagers Lab. Is a co-inventor on patents that include the use of AAVs for research and therapeutic applications (application numbers: 17/614,327, 63/332,655, 17/743,444, 63/344,328, 63/388,920). The other authors declare that no competing interests exist.

### Funding

| Funder | Grant reference number | Author |
|---|---|---|
| Templeton World Charity Foundation | 61350 | Vera Rinaldi |
| National Institutes of Health | R01HD080224 | Kathleen Desevin<br>Fengyun Sun<br>Oliver J Rando |
| National Institutes of Health | R01DA047678 | Andrew R Tapper |
| National Institutes of Health | DP1AG063419 | Kathleen Messemer<br>Amy J Wagers |

The funders had no role in study design, data collection and interpretation, or the decision to submit the work for publication.

### Author contributions

Vera Rinaldi, Conceptualization, Data curation, Formal analysis, Validation, Investigation, Visualization, Methodology, Writing – original draft, Project administration, Writing – review and editing; Kathleen Messemer, Kathleen Desevin, Fengyun Sun, Bethany C Berry, Shweta Kukreja, Investigation; Andrew R Tapper, Resources, Supervision, Methodology; Amy J Wagers, Supervision, Funding acquisition, Investigation, Project administration; Oliver J Rando, Conceptualization, Resources, Data curation, Formal analysis, Supervision, Funding acquisition, Visualization, Writing – original draft, Project administration, Writing – review and editing

### Author ORCIDs

Amy J Wagers http://orcid.org/0000-0002-0066-0485
Oliver J Rando http://orcid.org/0000-0003-1516-9397

### Ethics

All animal use was approved by the Institutional Animal Care and Use Committees of UMass Chan Medical School and Harvard Faculty of Arts and Sciences, under protocol PROTO202100029 to Dr. Oliver Rando and protocol 29014 awarded to Dr. Amy Wagers.

### Decision letter and Author response

Decision letter https://doi.org/10.7554/eLife.77733.sa1
Author response https://doi.org/10.7554/eLife.77733.sa2

---

## Additional files

### Supplementary files

• Supplementary file 1. Oligonucleotides used in this study. Table lists oligo sequences used for genotyping PCRs.

• Supplementary file 2. RNA-Seq from FACS-isolated Tomato-positive cells. Table shows mRNA abundance (expressed as FPKM) for RNA-Seq libraries obtained from FACS-isolated Tomato-positive cells obtained from the cauda epididymis of Ai14D animals carrying either *Defb41^{Cre}* or *Slc32a1^{Cre}*, as indicated.

• Transparent reporting form

• Source data 1. All source data files are original uncropped gel images used in the corresponding figures: *Figure 3E*, *Figure 5C*, *Figure 6C*, *Figure 6F*, *Figure 4—figure supplement 1C*, *Figure 6—figure supplement 1C*, *Figure 6—figure supplement 1F*, *Figure 6—figure supplement 2*.

### Data availability

All data generated or analysed during this study are included in the manuscript, supplementary and source data files. Raw image data is available at *BioStudies* under accesscion number S-BIAD668.

The following dataset was generated:

| Author(s) | Year | Dataset title | Dataset URL | Database and Identifier |
|---|---|---|---|---|
| Rando O | 2023 | Evidence for RNA or protein transport from somatic tissues to the male reproductive tract in mouse | https://www.ebi.ac.uk/biostudies/bioimages/studies/S-BIAD668 | BioStudies, S-BIAD668 |

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
