## [Editor Report]

This manuscript reports data consistent with a new and unanticipated phenomenon: that Cre or its mRNA may be transmitted between tissues in the mouse and that the male reproductive tract (epididymis) appears to be the most common target of such transported molecules. The data serve as a timely warning to mouse researchers about an unexpected complication of Cre-mediated gene manipulation.

---

## [Decision Letter]

**Decision letter after peer review:**

Thank you for submitting your article "Evidence for RNA or protein transport from somatic tissues to the male reproductive tract in mouse" for consideration by *eLife*. Your article has been reviewed by 3 peer reviewers, and the evaluation has been overseen by a Reviewing Editor and Didier Stainier as the Senior Editor. The following individual involved in review of your submission has agreed to reveal their identity: Terence Pang (Reviewer #2).

Essential revisions:

It is recommended that the revisions include: (1) changes to text to temper claims and (2) consideration of additional experimental data that may help support the author's conclusions:

1) Please explain/consider:

(i) how the non-uniform transfer of Cre among epididymal cells (particularly in the caput segment) fit with the mechanistic explanation of the epididymis as a "trash can"?

(ii) the age of the animals used in the initial experiments (Figure 1) and whether age / sex steroid production plays any role in regulating Cre transfer/activity.

(iii) why an exposure regimen of 3 transfers over 5 days was selected.

2) Please consider:

(i) injection of Cre, or Cre-encoding RNA in liposomes or artificial exosomes into the tail of the mice, prior to testing for epididymal editing.

(ii) the use of an alternative Cre-dependent system (e.g. using a Cre-DREDD) to eliminate the prospect that this is an artifact associated with the use of tdTOMATO.

(iii) the possibility of performing PCR quantification to detect AAV viral particles in the caput.

(iv) examining the expression of the "brain tissue-specific" Cre driver gene in the epididymis in wildtype mice and the Cre lines to confirm tissue specificity and eliminate the possibility of transcriptional leakage.

(v) please check PCR conditions and DNA quality to address several of the comments raised by Referee 3 (i.e. comments 2, 3, 4, 6 and 7).

*Reviewer #1 (Recommendations for the authors):*

1. It is interesting that the Cre does not appear to transfer between cells in the epididymis, that the edited areas are patchy in some cases. Similarly not all Cre lines seemed to send Cre to the same epididymal cells. How does that fit mechanisms like "trash can"?

2. Considering the authors' hypothesis and their test using serum or exosome injection, why not test the model by injecting Cre, or Cre-encoding RNA in liposomes or artificial exosomes into the tail of the mice, and test for epididymal editing?

3. In addition to having clear and convincing figures, the paper is well written, with the authors anticipating questions that could arise, and providing experiments or arguments to address them. An example is their compelling argument against the idea that Cre is transmitted directly by neurons.

*Reviewer #2 (Recommendations for the authors):*

If off-target expression is noted in the male reproductive tract, is it also detected in the female reproductive tract? Addressing this might reveal clues to the specificity of this present observation, perhaps tracing back to the primordial divergence in male/female reproductive organs The epididymis develops from the mesonephric duct and while development is regressed in females, there may be remnant tissue/cells which also express Cre.

The other commonality to all the experiments is the use of tdTOMATO. An alternative Cre-dependent system should be explored e.g. using a Cre-DREDD?

The age of the animals used in the initial experiments (Figure 1) is not clearly stated in the Methods, or figure legends. I can only assume these are 8-12 week old mice as that is the age then the subsequent AAV injection study was conducted. Is Cre activity similarly expressed in young juvenile animals? If not, perhaps there is some level of regulation by sex steroids?

What is the basis for the differential expression of Cre within the caput epididymis itself? This is quite the outstanding visual difference in expression and should be considered further. Could this be related to its pattern of venous perfusion? Or is it a more complex interaction between the CLARITY process and the cellular/structural composition of this region?

Was confirmatory profiling of neuron-specific genes and adipose performed on the caput vs corpus/cauda to confirm their presence/expression? The diffuse pattern of expression is perhaps suggestive of some level of structural specificity. Does this align with the authors' previous work on the epididymal cell atlas?

With regards to the AAV study, is it possible to perform PCR quantification to detect viral particles in the caput? Also, in this case, it would be interesting to know how long is Cre expression maintained for?

It is not clear why a limited exposure of three transfers over 5 days was selected. Was this protocol based on published methodology, including administration through the tail vein? Perhaps the inconsistent outcomes were due to a larger exposure being required.

*Reviewer #3 (Recommendations for the authors):*

I suggest the authors to spend more time to make their data more complete and thus conclusive, as to make the result stand the test of time.

1. First and foremost, the author should check the expression of the "brain tissue-specific" Cre driver gene in the epididymis (e.g., mRNA level) in wildtype mice and the Cre lines, because the tissue specificity of these genes may not be so specific. For example, the pan-neuronal maker Syn1 are also identified expressed in the human sperm [PMID: 26566474]. A stochastic expression of the Cre driver gene can result in the expression of Cre protein that are sufficient to induce cleavage in the expressed few cells, this is consistent with authors observation that the Cre-loxp recombination effect are shown in cell scattered in the epididymal/vas deferens tissue. My point is, this possibility of transcriptional leakage should be stringently tested, which could be a main reason of the Cre leakage, or exist in paradelle to the other proposed reason (Cre mRNA/protein transported from another tissue).

2. Figure 3: In the E panel, there is no PCR product of Vas def under the No Cre condition both for the Rec 7 and Rec 11. This is not correct. Without Cre, there should be a larger intact band as observed in other samples. Similar problems have been observed in Figure 5c, where serval samples were failed to detect a PCR product including Sem Ves, Kidney. The author should re-check their PCR condition or the DNA quality of these tissues. This type of data, in my opinion, is not suitable for publication because this lead to doubts on the reliability of the PCR result, and decrease the credibility of the conclusion.

3. Also, in Figure 3B panel, the author shows the image of liver, kidney, intestine, and testis, it is better to including the PCR results of these tissues in the E panel, which will make the data more consistent (gross image showing on red fluorescence signal is a weak demonstration, because the tissue are pretty large and there could be scattered red signal in a few cells can could only be observed under tissue section).

4. Moreover, in Fig3E, the author should clarify the PCR product length of the original DNA and recombined DNA by the Rec7 and Rec11 primers in their Figure legends, some reader may neglect the primer details in the method section. Did the author wrongly label the Rec11 and Rec7? since the Rec11 should have a larger length for the recombined PCR product, but in the figure it shows a smaller product. Also, there is faint band in the H2O Lane for the Rec11 primer which may cause false positive in this assay in other experiments, the author should be more careful to interpretate the results with faint band observed.

5. For the parabiotic assay, did the author check other tissues in the Ai14D recipient, at least, brain tissue should be included in the check list besides the epididymis, since the Vgat-cre is primarily target for cell in the brain tissue.

6. In the Figure S9E, in the "negative" group, there are still some red fluorescence observable in the photos. Did the author check the negative samples by the PCR assay?

7. It would be better to label or clarify which primer were used to check the recombination events in the Figure 5C, 6C, 6F et al.

---

## [Author Response]

Essential revisions:It is recommended that the revisions include: (1) changes to text to temper claims and (2) consideration of additional experimental data that may help support the author's conclusions:1) Please explain/consider:(i) how the non-uniform transfer of Cre among epididymal cells (particularly in the caput segment) fit with the mechanistic explanation of the epididymis as a "trash can"?

The “trash can” model is of course not the model we favor here, it is simply an abundance of caution that leads us to suggest this alternative explanation. As for the heterogeneity in recipient cells, the same arguments could be made in the trash can model as in the directed trafficking model – in the latter, one envisions only a subset of cells expressing some tetraspanin or other exosome receptor, while in the former one can imagine only a subset of cells being responsible for clean up of circulating material (or being located near leaky vasculature, etc.). So we do not think the patchy Cre reception distinguishes between the models.

(ii) the age of the animals used in the initial experiments (Figure 1) and whether age / sex steroid production plays any role in regulating Cre transfer/activity.

This is an interesting point and one we have not explored in great detail. Basically, every animal assayed in this study was taken relatively soon after male mice become sexually mature – double transgenic and AAV recipient animals were typically sacrificed for assays ~10-12 weeks after birth. For serum transfers, donors ranged from 14 up to 45 weeks old, while recipients started receiving injections around 12 weeks of age and were sacrificed a few weeks later.

(iii) why an exposure regimen of 3 transfers over 5 days was selected.

We actually continuously modified our transfer protocols over tens of experiments in an (unsuccessful in the event) attempt to identify conditions that consistently yielded reliable reporter activity in the recipient animal. However, our initial experiments – which lead to successful reporter activity in the recipient – and indeed most of those reported in Figures 6 and S9 (now Figure 6-supplement 1), used this transfer scheme. In other words, we chose this regimen because it worked initially. However, the variable success in subsequent trials led us to add more transfer days, or increase the amount of material transferred, but as detailed in the manuscript we were never able to find an exposure regime that reliably leads to recombination in the recipient animal. This is of course frustrating but it is a fact, and is the reason for the heavily guarded and cautious language used throughout the manuscript.

2) Please consider:(i) injection of Cre, or Cre-encoding RNA in liposomes or artificial exosomes into the tail of the mice, prior to testing for epididymal editing.

This is an interesting idea, and one we had been considering as a complement to the exosome transfers from “donor” animals. However, as noted the efficiency of this process, using circulating material from Cre-expressing animals, is extremely poor. Given that any liposome or artificial exosome-based experiment will have the added issue that such injected material would not have any theoretical targeting protein (eg exosome surface proteins – tetraspanins most likely) that target the epididymis, we chose not to pursue this approach as it seems a priori far less likely to be successful.

(ii) the use of an alternative Cre-dependent system (e.g. using a Cre-DREDD) to eliminate the prospect that this is an artifact associated with the use of tdTOMATO.

This is an excellent and very important suggestion, thank you. As noted above, this has been the primary focus of our experimental approach to revising the manuscript. Gratifyingly, we find that *AdipoQ^Cre^* drives recombination of another target locus – a floxed allele of *Dcr1* – in the cauda epididymis (new Figure 5-supplement 1). This greatly reduces any concerns that our results reflect a somehow unusual target locus in the Ai14D reporter, and gives us greater confidence in our core finding.

(iii) the possibility of performing PCR quantification to detect AAV viral particles in the caput.

This would have been possible but we do not have any material remaining from the AAV-injected animals. Also, the fact that recombination occurs in double transgenic animals, and in parabiosis and exosome transfer recipients, both make this control somewhat superfluous in our view.

(iv) examining the expression of the "brain tissue-specific" Cre driver gene in the epididymis in wildtype mice and the Cre lines to confirm tissue specificity and eliminate the possibility of transcriptional leakage.

This is a very important point, which we did originally address albeit with a single sentence. We have extensive RNA-Seq data from the epididymis, with bulk RNA-Seq of the caput, corpus, cauda, and vas, as well as a single-cell RNA-Seq atlas (Rinaldi et al. 2020). We did interrogate this dataset for expression of the genes associated with our various Cre drivers: “We note that the majority of these genes are either undetectable or expressed at extremely low levels (~1 ppm) in the epididymis as assayed either by bulk or single cell RNA-Seq”. As this is indeed an important point, to emphasize it we have now also reiterated this point in the revised Discussion.

(v) please check PCR conditions and DNA quality to address several of the comments raised by Referee 3 (i.e. comments 2, 3, 4, 6 and 7).

As requested, we isolated fresh genomic DNA and repeated the genotyping PCRs in Figure 3E. We now have the expected negative bands in the spleen and vas deferens samples that were missing previously.

Reviewer #1 (Recommendations for the authors):1. It is interesting that the Cre does not appear to transfer between cells in the epididymis, that the edited areas are patchy in some cases. Similarly not all Cre lines seemed to send Cre to the same epididymal cells. How does that fit mechanisms like "trash can"?

The “trash can” model is of course not the model we favor here, it is simply an abundance of caution that leads us to suggest this alternative explanation. As for the heterogeneity in recipient cells, the same arguments could be made in the trash can model as in the directed trafficking model – in the latter, one envisions only a subset of cells expressing some tetraspanin or other exosome receptor, while in the former one can imagine only a subset of cells being responsible for clean up of circulating material. So we do not think the patchy Cre reception distinguishes between the models.

2. Considering the authors' hypothesis and their test using serum or exosome injection, why not test the model by injecting Cre, or Cre-encoding RNA in liposomes or artificial exosomes into the tail of the mice, and test for epididymal editing?

This is an interesting idea, and one we had been considering as a complement to the exosome transfers from “donor” animals. However, as noted the efficiency of this process, using circulating material from Cre-expressing animals, is extremely poor. Given that any liposome or artificial exosome-based experiment will have the added issue that such injected material will not have any theoretical targeting protein (eg exosome surface proteins – tetraspanins most likely) that target the epididymis, we chose not to pursue this approach as it seems a priori far less likely to be successful.

3. In addition to having clear and convincing figures, the paper is well written, with the authors anticipating questions that could arise, and providing experiments or arguments to address them. An example is their compelling argument against the idea that Cre is transmitted directly by neurons.

We thank the reviewer for the kind and supportive comments.

Reviewer #2 (Recommendations for the authors):If off-target expression is noted in the male reproductive tract, is it also detected in the female reproductive tract? Addressing this might reveal clues to the specificity of this present observation, perhaps tracing back to the primordial divergence in male/female reproductive organs The epididymis develops from the mesonephric duct and while development is regressed in females, there may be remnant tissue/cells which also express Cre.

This is an interesting question, and one we did look into, albeit shallowly. We have found little evidence for Tomato expression – one fleck of fluorescence in lightsheet images, or a single cell across multiple histology sections – in the ovary, uterus or oviducts in AAV recipients, or in double transgenics carrying Vgat, Alb, Fev, ChAT Cre drivers. We did observe some Tomato signal in the uterus and oviducts of Ai14D;*Nest^Cre^* double transgenics, but did not characterize this signal in any detail. So overall we find much less evidence for this type of trafficking in the female reproductive tract, but we did not survey enough animals to make this claim with any confidence.

The other commonality to all the experiments is the use of tdTOMATO. An alternative Cre-dependent system should be explored e.g. using a Cre-DREDD?

This is an excellent suggestion, thank you. As noted above, this has been the primary focus of our experimental approach to revising the manuscript. Gratifyingly, we find that *AdipoQ^Cre^* drives recombination of another target locus – a floxed allele of *Dcr1* – in the cauda epididymis (new Figure 5-supplement 1). This greatly reduces any concerns that our results reflect a somehow unusual target locus in the Ai14D reporter, and gives us greater confidence in our core finding.

The age of the animals used in the initial experiments (Figure 1) is not clearly stated in the Methods, or figure legends. I can only assume these are 8-12 week old mice as that is the age then the subsequent AAV injection study was conducted. Is Cre activity similarly expressed in young juvenile animals? If not, perhaps there is some level of regulation by sex steroids?

Apologies, this should have been detailed, thanks for pointing this out. We have now clarified the ages of the animals used in the Methods section. As the reviewer speculates, we typically use 8-12 week old males for AAV experiments and double transgenics. The only exceptions were serum/exosome transfers, where recipients were typically ~10-15 weeks of age, but donors were usually 15-25 weeks of age as this allowed for more serum to be obtained.

We have not done any systematic analysis of Cre activity over an animal’s lifetime, although nothing in our experience suggested more reporter activity in older animals than in younger ones. It is an interesting possibility but we did not explore this systematically.

What is the basis for the differential expression of Cre within the caput epididymis itself? This is quite the outstanding visual difference in expression and should be considered further. Could this be related to its pattern of venous perfusion? Or is it a more complex interaction between the CLARITY process and the cellular/structural composition of this region?

I am not 100% sure about the meaning of this question. Is the question why some parts of the overall epididymis tend to light up? Most Cre drivers light up the cauda – distal – epididymis, with less activity in the caput. Or if the question is read literally, the question is why, within one section of the epididymis, Tomato expression is patchy/heterogeneous. To some extent we suspect the answer to both questions could be variability in the distribution of, say, an exosome receptor on cell surfaces – some tetraspanin most likely. So we might see better targeting of the cauda over the caput thanks to *Tspan8* expression in the cauda epithelium but little expression in the caput. Similarly, cell to cell variability within a segment in *Tspan8* (etc.) expression could be the reason for the patchy reporter activity. Beyond that, the two questions could also have distinct answers. Targeting of the cauda but not the caput for *Vgat^Cre^* for example could conceivably be driven by differences in the structure and/or penetration of vasculature into this tissue. The caput epididymis, for instance, is characterized by fenestrated endothelium and as such would be more likely to be targeted nonspecifically by trafficked vesicles – comfortingly, we tend to see better targeting of the cauda than the caput, so we are not particularly concerned that the location of fenestrated endothelium explains our findings.

Was confirmatory profiling of neuron-specific genes and adipose performed on the caput vs corpus/cauda to confirm their presence/expression? The diffuse pattern of expression is perhaps suggestive of some level of structural specificity. Does this align with the authors' previous work on the epididymal cell atlas?

This is a good question. The answer is that by and large we do not see expression of adipose and neuron-specific genes in the epididymis, either in bulk tissue RNA-Seq or in our single cell atlas. There are of course overlapping genes – *Npy* (encoding neuropeptide Y) is an unexpected marker of Segment 7 Principal Cells – but none of the promoters we explored in this study (eg *Vgat*, *AdipoQ*, etc.) is associated with anything beyond minimal (~1 ppm or less) mRNA levels in the epididymis. There are two likely explanations here. One would be that exosomal RNA packaging has some specificity for a subset of mRNAs in the originating cell, and *Cre* turns out to be a good candidate for inter-tissue trafficking while *Vgat* and other genes are not as enriched or are depleted from circulating information. Whether or not this is true, the other key point is that Cre-Lox genetics has a very high gain – one molecule of Cre could in principle be sufficient to drive LoxP recombination, leading to permanent Tomato expression well after the degradation of the instigating Cre molecule. So we believe the incredibly high signal amplification possible with Cre-Lox allows us to observe evidence for Cre activity, while rare molecules of *Vgat* mRNA (etc.) will not appreciably affect the RNA landscape of epididymal tissue.

With regards to the AAV study, is it possible to perform PCR quantification to detect viral particles in the caput? Also, in this case, it would be interesting to know how long is Cre expression maintained for?

This would have been possible but we do not have any material remaining from the AAV-injected animals. Also, the facts that recombination occurs in double transgenic animals, and in parabiosis and exosome transfer recipients, both make this control somewhat superfluous in our view.

It is not clear why a limited exposure of three transfers over 5 days was selected. Was this protocol based on published methodology, including administration through the tail vein? Perhaps the inconsistent outcomes were due to a larger exposure being required.

We actually continuously modified our transfer protocols over tens of experiments in an (unsuccessful in the event) attempt to identify conditions that consistently yielded reliable reporter activity in the recipient animal. However, our initial experiments – which lead to successful reporter activity in the recipient – and indeed most of those reported in Figures 6 and S9 (new Figure 6-supplement 1), used this transfer scheme. In other words, we chose this regimen because it worked initially. However, the variable success in subsequent trials led us to add more transfer days, or increase the amount of material transferred, but as detailed in the manuscript we were never able to find an exposure regime that reliably leads to recombination in the recipient animal. This is of course frustrating but it is a fact, and is obviously the reason for the guarded and cautious language used throughout the manuscript.

Reviewer #3 (Recommendations for the authors):I suggest the authors to spend more time to make their data more complete and thus conclusive, as to make the result stand the test of time.1. First and foremost, the author should check the expression of the "brain tissue-specific" Cre driver gene in the epididymis (e.g., mRNA level) in wildtype mice and the Cre lines, because the tissue specificity of these genes may not be so specific. For example, the pan-neuronal maker Syn1 are also identified expressed in the human sperm [PMID: 26566474]. A stochastic expression of the Cre driver gene can result in the expression of Cre protein that are sufficient to induce cleavage in the expressed few cells, this is consistent with authors observation that the Cre-loxp recombination effect are shown in cell scattered in the epididymal/vas deferens tissue. My point is, this possibility of transcriptional leakage should be stringently tested, which could be a main reason of the Cre leakage, or exist in paradelle to the other proposed reason (Cre mRNA/protein transported from another tissue).

Reviewer 2 also asked this question, which is a good one. As detailed in our response above, the majority of the genes used as Cre drivers here are not detectably expressed in either bulk RNA-Seq or in single cell RNA-Seq atlases. If our data result from leaky expression it must be of extraordinarily low level. Of course with the huge signal amplification afforded by Cre-Lox recombination this could still drive Tomato expression. Therefore our strongest argument against leaky Cre expression directly in the cells of interest can only be the AAV results, and the parabiosis and exosome transfer data (as inefficient as they may be). But to address the specific request here: Vgat, AdipoQ, and several of the other drivers are not associated with detectable mRNAs in the epididymis.

2. Figure 3: In the E panel, there is no PCR product of Vas def under the No Cre condition both for the Rec 7 and Rec 11. This is not correct. Without Cre, there should be a larger intact band as observed in other samples. Similar problems have been observed in Figure 5c, where serval samples were failed to detect a PCR product including Sem Ves, Kidney. The author should re-check their PCR condition or the DNA quality of these tissues. This type of data, in my opinion, is not suitable for publication because this lead to doubts on the reliability of the PCR result, and decrease the credibility of the conclusion.

As requested, we have isolated fresh genomic DNA and repeated these genotyping PCRs in the revised Figure 3E. We now observe the expected negative bands in the vas deferens and spleen samples.

3. Also, in Figure 3B panel, the author shows the image of liver, kidney, intestine, and testis, it is better to including the PCR results of these tissues in the E panel, which will make the data more consistent (gross image showing on red fluorescence signal is a weak demonstration, because the tissue are pretty large and there could be scattered red signal in a few cells can could only be observed under tissue section).

Unfortunately, we did not have any extra collected samples of these tissues archived for genomic DNA extraction – we typically would only collect a few mm^3^ of any given tissue for molecular analysis – and could not perform these PCRs.

4. Moreover, in Fig3E, the author should clarify the PCR product length of the original DNA and recombined DNA by the Rec7 and Rec11 primers in their Figure legends, some reader may neglect the primer details in the method section. Did the author wrongly label the Rec11 and Rec7? since the Rec11 should have a larger length for the recombined PCR product, but in the figure it shows a smaller product. Also, there is faint band in the H2O Lane for the Rec11 primer which may cause false positive in this assay in other experiments, the author should be more careful to interpretate the results with faint band observed.

We have added the PCR product lengths in the Figure legends as requested. As far as switching the labels, the reviewer may be correct about the original figure – in any case, in the revised PCR it is clear from the markers in Figure 3E that the Rec7 band lies between the 200 and 300 bp marker bands, while the Rec11 band lies between the 300 and 400 bp markers, as expected. In addition, in the repeated PCRs we do not see the band noted in the H2O lane.

5. For the parabiotic assay, did the author check other tissues in the Ai14D recipient, at least, brain tissue should be included in the check list besides the epididymis, since the Vgat-cre is primarily target for cell in the brain tissue.

While this is an interesting experiment, we are unable to perform the requested analysis as we do not have any remaining tissue for further analysis.

6. In the Figure S9E, in the "negative" group, there are still some red fluorescence observable in the photos. Did the author check the negative samples by the PCR assay?

We did not – sample preparation for lightsheet imaging is incompatible with the PCR assay so the same sample cannot be used for both assays.

7. It would be better to label or clarify which primer were used to check the recombination events in the Figure 5C, 6C, 6F et al.

Fixed as requested.